# Self-Organization Effects of Thin ZnO Layers on the Surface of Porous Silicon by Formation of Energetically Stable Nanostructures

**DOI:** 10.3390/ma16020838

**Published:** 2023-01-15

**Authors:** Danatbek Murzalinov, Ainagul Kemelbekova, Tatyana Seredavina, Yulia Spivak, Abay Serikkanov, Aigul Shongalova, Sultan Zhantuarov, Vyacheslav Moshnikov, Daniya Mukhamedshina

**Affiliations:** 1Institute of Physics and Technology, Satbayev University, Almaty 050013, Kazakhstan; 2Microelectronics Department, Saint-Petersburg State Electrotechnical University, 5 Professora Popova Street, 197376 Saint-Petersburg, Russia

**Keywords:** paramagnetic particles, saturation of the EPR signal, pore hierarchy, nanoclusters, ZnO, photoluminescence, relaxation time

## Abstract

The formation of complex surface morphology of a multilayer structure, the processes of which are based on quantum phenomena, is a promising domain of the research. A hierarchy of pore of various sizes was determined in the initial sample of porous silicon by the atomic force microscopy. After film deposition by spray pyrolysis, ZnO nanoclusters regularly distributed over the sample surface were formed. Using the electron paramagnetic resonance (EPR) method it was determined that the localization of paramagnetic centers occurs more efficiently as a result of the ZnO deposition. An increase in the number of deposited layers, leads to a decrease in the paramagnetic center relaxation time, which is probably connected with the formation of ZnO nanocrystals with energetically stable properties. The nucleation and formation of nanocrystals is associated with the interaction of particles with an uncompensated charge. There is no single approach to determine the mechanism of this process. By the EPR method supplemented with the signal cyclic saturation, spectral manifestations from individual centers were effectively separated. Based on electron paramagnetic resonance and photoluminescence studies it was revealed that the main transitions between energy levels are due to oxygen vacancies and excitons.

## 1. Introduction

Composites based on nanomaterials embedded in matrices of various natures are promising for new-generation functional devices. They can consist of a wide range of compounds comprising inorganic, organic, and hybrid structures [1].

The self-organization effects of low-dimensional system on the surface of a solid substance through the formation of periodically ordered structures are of great interest. It is related to the fact that the process is determined by quantum phenomena, suggesting novel approaches to the comprehension of matter formation nature.

Porous silicon (por-Si) is an attractive material due to the fact that its internal volume can be used as nanoreactor for the synthesis of various particles. Spatial pore separation reduces the effect of nanoparticle aggregation. Controlling the shape and size of the channels, it is possible to explore various materials with specified geometric dimensions and shapes. Changing the surface morphology—sizes, the arrangement uniformity of pore outlets, and the surface roughness—it is possible to control the process of nucleation during the synthesis of a substance on its surface. In this case, the composition of surface adsorption centers, its energy characteristics, and wettability properties play a significant role. Moreover, por-Si is a promising material due to its obvious compatibility with silicon microelectronics [2,3].

Structures based on zinc oxide particles integrated into silicon substrates are promising as components of various semiconductor devices. The formation of ZnO nanoclusters on the surface and in the pores of a sample is an important process that can be used in gas sensors, as their sensitivity enhances with increasing surface area [4,5,6,7,8,9,10,11,12,13].

Due to their unique properties, II–VI group semiconductors are widely used in solar cells, field effect transistors, optoelectronic devices, diluted magnetic semiconductors (DMS), photoluminescence appliances [14,15,16]. Owing to a high exciton binding energy (60 meV) and a wide band gap (3.4 eV), ZnO is one of the most useful among them. Therefore, zinc oxide is one of the most promising photonic materials in blue-ultraviolet (UV) region [17].

ZnO involves a wurtzite structure with lattice constants a = 0.32 nm and c = 5.52 nm. It belongs to the space group C6_3_mc and point group 6mm. Tetrahedrally coordinated O^2−^ and Zn^2+^ ions form the ZnO structure. The 4s electrons of the outer orbital of the Zn ion form the conduction band, yet the electrons of the 2p orbital of the O ion form the valence band. The intraband and interband levels in the electronic structure of ZnO are formed by the bonding of outer orbitals with defects [18].

One of the most essential properties of ZnO is its powerful interaction with light, leading to photoinduced effects. This effect is mainly conditioned to the properties of excitons and point defects. The high binding energy of excitons allows to exhibit effective luminescence in the near UV range even at room temperature and to transmit 80–90% of the light in the visible range [19].

The nature of ZnO magnetic and optical properties is associated with the defect presence in the structure. One of the frequently observed EPR signals in ZnO with an isotropic g-factor of ~1.955–1.964 is attributed to a singly charged oxygen vacancy, which can be compared to an F center. Three degenerate states of the particle exist: charged state +2, charged state +1, and neutral state. Meanwhile, high-field EPR investigations of ZnO identified axial g-anisotropy. This indicates the presence of several types of particles with an uncompensated charge in the structure [20].

The morphology of the sample surface depends on the defect configuration at the nanoscale due to their selective localization (nanoparticles, nanosheets, nanorods, nanoshells, and quantum dots).

The results of the energy distribution analysis of adsorption surface centers carried out by the Tanabe method [21] and the EPR data complement each other.

The sol–gel method is promising for the formation of chemically active structures. The reaction rate in this method is easily controlled by changing the process temperature. It is possible to influence the growth rate in a certain crystallographic direction. It allows to obtain thin ZnO films with preferential orientation in the structure [22,23], and to apply new methods for the formation of nanopatterns [24,25].

These techniques provided a combination of selectivity and high sensitivity for creating multisensory systems [26].

The spray pyrolysis method involves several advantages over various thin film deposition methods, such as high deposition rate, stoichiometry control, and facile coverage of large surface areas. It is a comprehensible, inexpensive, and vacuum-free synthesis method. As it is based on solution chemistry processes, dopants can be effortlessly introduced into the main structure [27].

The process of forming materials with high luminescent and sensory properties is of great interest. The nucleation of structures occurs at the level of interaction of individual particles, with an uncompensated charge. However, the isolation of the photoluminescence and EPR spectroscopic signal of such particles is an extremely difficult task.

Nanocrystalline structures have high luminescent characteristics and sensitivity when reacting with gases. *Nanocrystal–amorphous* substance transitions have the structure of a gradual transition from formations of one type to another and therefore contain particles with dangling bonds that differ in structure. The key to understanding the mechanism of transition is related to the nature of the paramagnetic centers (PMCs) interaction.

During the formation of PMCs, their structure and energy properties change. The transformation of the PMC is limited in time, that is, one of the main characteristics is the relaxation time of the particles [28].

Therefore, the study of the EPR signal saturation, based on the dependence of the properties of the PMC on the relaxation time, will allow us to identify the process of nucleation of nanostructures and the mechanism of their formation.

In [29] ZnO nanoparticles were synthesized via a solid coprecipitation approach followed by a high intensity ball milling procedure. The study of microwave saturation is the effective way to investigate the dynamic properties of the two different defect paramagnetic centers. The surface signal has more pronounced saturation behavior than the bulk signal. Different mobilities of the corresponding electrons at the defect sites is the reason of this effect. The saturation study was carried out in the range of microwave power from 0.1 to 10 mW. By dividing the EPR spectra into power intervals and interpretation on several spectra makes it possible to isolate signals from individual centers.

Savchenko et.al. [19] consider the role of the paramagnetic donor-like defects in the high n-type conductivity of the hydrogenated ZnO particles. The EPR spectrum of ZnO microparticles is a superposition of the four components measured with the different microwave (MW) power levels varied in the range from 0.015 to 15 mW. However, it is impossible to determine anisotropy from the spectrum, which is associated with the presence of several signals. Probably the division into intervals of 0.015 to 15 mW would allow identifying anisotropy and isolating the signals of individual PMCs.

In [30], EPR spectroscopy of different sol concentrations synthesized nanocrystalline ZnO thin films. It is demonstrated that electrons in bulk ZnO are delocalized at the defect centers and it is difficult to saturate transitions at room temperature, and their EPR spectra are usually recorded at low temperatures [31].

The importance of these studies lies in a new approach to studying the EPR signal saturation process by stepwise changing the microwave power values at certain variable intervals. This made it possible to represent the change more clearly in the shape and component of the signal. The representation of signals in integral form with decomposition into components is also used for the purpose of identifying signals of different PMCs.

The aim of the work was to investigate the transformation of the surface morphology during the formation of por-Si/ZnO structures by the synthesis of light-emitting particles ZnO with an uncompensated charge.

The novelty of this study lies in the determination of the process of nanoclusters formation in por-Si/ZnO structures by the EPR signal saturation method, based on the dependence on the relaxation time of the paramagnetic center.

A substantial feature of this work is the film deposition on a surface with complex morphology, including meso and macro pores.

## 2. Materials and Methods

### 2.1. Materials and Synthesis

#### 2.1.1. Synthesis of Porous Silicon

Por-Si layers were obtained by electrochemical anodic etching of p-Si (100) with a resistivity of 12 Ω·sm, dopant impurity-B [32,33,34]. An aqueous-alcohol solution of hydrogen fluoride was used as the electrolyte. Hydrofluoric acid 45.00%, ST-10484, GOST 10484-78, CAS: 7664-39-3. Isopropanol SSPIRT-9805.F01080, GOST 9805-84, CAS: 67-63-0. The anodizing current density was 10 mA/cm^2^ for 10 min. Electrochemical anode etching was carried out in a single-chamber electrochemical cell. It was shown in [35,36] that a heterophase micro-mesoporous skin layer of complex composition can form on the surface of macroporous silicon. During electrochemical etching, this layer was formed as a result of the strength redistribution of electric field over the surface of growing porous Si. These processes involve the removal of reaction products and redeposition on the surface. In this work, the skin layer was removed chemically in a 20% aqueous solution of hydrogen fluoride for 1–2 min in order to reveal access for macropores. Immediately after chemical treatment, porous silicon samples were used to synthesize ZnO.

#### 2.1.2. Synthesis and Deposition of a ZnO Coating on a Substrate

##### Sol-Gel Method

The sol–gel method was applied to prepare the film-forming solution. The sol solution was obtained from zinc acetate dihydrate (Zn(CH_3_COO)_2_·2H2O) with a concentration of 0.1 M, mixed with 9 mL isopropanol (C_3_H_7_OH) as a solvent with the addition of 1 mL monoethanolamine (C_2_H_7_NO) as a stabilizing agent. The solutions used were prepared on the LAB-PU-01 orbital rotation shaker (rotation speed 150 rpm). The dissolution was carried out both at room temperature and when heated to 50 °C, which was a significant condition for the homogenization of the resulting solutions. The time of complete preparation of the solution was 1 h. Deposition of ZnO films on the substrate began no later than 1 h after the preparation of solutions.

The nucleation process plays a key role in the formation of microstructure of the resulting films. It is especially essential for growing thin films with nanometer thickness.

##### The Process of Nucleation Using the Spin-Coating Method

A uniform sol layer was obtained by depositing a few drops of the solution onto a substrate fixed on a horizontal table, followed by table rotation at a speed of ~3400 rpm for 1 min. In this case, only such a layer remained on the surface that could be held by surface tension forces, and all excess solution was removed. After that, the substrates were dried at a temperature of 130 °C, followed by annealing at a temperature of 450 °C (1 h), which formed ZnO films on the substrates. This technique ensured the uniform growth of ZnO over the entire surface of the substrate.

##### Deposition of the Main Film Layers

On the heating element, there are substrates on which liquid solutions are sprayed using a pneumatic airbrush. The carrier gas (air) is supplied by a compressor through a filter and pressure regulator. The distance from the airbrush nozzle to the substrates varied in the range of 20–30 cm. The pressure was set to 1.4 bar, and the most uniform spray solution flow was observed.

The temperature range of the substrate was 350–400 °C. In this case, the solvent evaporated before the aerosol droplets reached the surface. Microcrystals were deposited on the substrate and decomposed on it. As a result, zinc oxide films with spherical and hexagonal crystallite shapes were synthesized [37].

The synthesis time for one layer was 30 s, and for twenty layers about 40 min. Further, the obtained samples were subjected to annealing for 15 min. Annealing between the deposition of individual layers was 15 s.

In these investigations, 20 and 25 coating layers were deposited, since the goal was to research thin, yet volumetrically formed structures.

### 2.2. Characterization Methods

The structure of the deposited films was studied by X-ray diffraction (XRD) with the DRON-6 (“Burevestnik”, Saint-Petersburg, Russia) setup. A narrow-collimated (0.05 × 1.5) mm^2^ of a monochromatic (CuKa) X-ray beam directed at an angle of 5° to the surface of the sample was used. The intensity of X-ray reflections along the debagram was measured every 2θ = 0.05 on the MD-100 microdensitometer.

The SORBTOMETER-M (“Catakon”, Novosibirsk, Russia) device is designed to determine the total specific surface area of porous substances and materials by thermal desorption of gas-adsorbate by the BET method. Specific surface measurement range—0.3–2000 m^2^/g. The temperature range of sample thermal training is 50–300 °C. 

Process of analyzing the specific surface area of samples included the following stages. Firstly, the samples were weighted. Further, the flask with the samples was heated to 150 °C in an argon atmosphere (adsorptive gas) for 20 min. The following step was to measure the specific surface area of the samples through adsorption and desorption of gases with repetitions in 1 cycle and subsequent analysis. The principle of the analyzer operation is based on the dynamic (thermal desorption) method, which consists in the measurements of the volume of the adsorbed argon, being in sorption equilibrium and in contact with a dispersed porous material. The results of measuring of the adsorbed gas volume are used to calculate the specific surface area based on the Brunauer–Emmett–Teller (BET) equation.

EPR spectrometer JES-FA200 (“JEOL”, Akishima, Japan). Measurements in the ranges ~9.4 GHz (X-Band) and ~35 GHz (Q-Band). Microwave frequency stability ~ 10^−6^. Sensitivity–7 × 10^9^/10^−4^ Tl. Resolution-2.35 μT. Output power—from 200 mW to 0.1 μW, quality factor (Q-factor) 18,000.

EPR spectra were recorded at room temperature with a JES-FA spectrometer (Jeol) in the presence of the standard Mn^2+^/MgO. Registration conditions: Frequency 9.445 GHz, in the field Fc = 340 mT, sweep width 7.50 (mT), modulation frequency and amplitude Fr. = 100 (kHz), width = 0.6 (mT).

When investigating the nature of EPR signal saturation, the spectrum was recorded under identical conditions and with a change in the microwave radiation power in the range from 0.4 to 7.6 mW. These investigations were carried out for samples with 20 layers (m = 4.1 mg) and with 25 layers (m = 7.8 mg) of coating. 

To isolate the EPR signal from the noise of the original spectra, the accumulation was carried out four times, and to investigate the saturation of the signal, the accumulation was carried out six times. In the second case, the number of accumulations is conditioned to the complexity of isolation of a gradually increasing signal.

Atomic force microscope (AFM) JSPM-5200 (“JEOL”, Akishima, Japan). Scanning mode (AFM AC), typical scanning speeds (the scanning time of 1 line is 625 microseconds, the scanning time of the entire image is 13 min), and brands of probes applied (NSC35/AIBS). Scan scale varied as follows—25 × 25 µm, 6 × 6 µm, 1.5 × 1.5 µm, 912 × 912 nm, 500 × 500 nm. Operating temperature range—from 100 °C to 500 °C, vacuum depth—up to 10^−6^ mm Hg. In addition, optical microscopy images were investigated, with a magnification of 125 times TM-24033 (“JEOL”, Akishima, Japan). 

Photoluminescence (PL) was measured using an Cary Eclipse (“Agilent”, Santa Clara, CA, USA) spectrophotometer in the spectral range from 200 to 800 nm. The spectral width of the slit for this device is variable and is on the order of 0.5–2.4 nm. The optical scheme of the spectrophotometer is based on a monochromator with a concave holographic diffraction grating having 1023 lines/mm. A tungsten-halogen lamp for measurements in the visible region of the spectrum and a deuterium lamp for UV measurements are applied as a radiation source. 

Spectral ellipsometer Ellipse-1891 (“Nanotechnology center”, Novosibirsk, Russia). The light source is a high-pressure xenon lamp. Spectral range—250–1000 nm. Single measurement time—5 ms/point, full spectrum—8–20 s. Light beam diameter—3 mm. The angle range of light incidence is 45, 50, 55, 60, 65, 70, and 90 degrees. Reproducibility—dψ = 0.02°, Dδ = 0.05°.

## 3. Results and Discussion

### 3.1. Surface Morphology Investigation

The surface structure of the samples was investigated by optical and atomic force microscopy. The choice of the minimum scale of 500 × 500 nm is due to the possibility of identifying objects with various dimensions.

ZnO particles with low surface tension are able to penetrate por-Si with precise concentrations of Zn^2+^.

The surface of the sample without film deposition, when magnifying using an optical microscope, demonstrates a macroporous structure (Figure 1a). The surface morphology is homogeneous. In the image of a scan area of 25 × 25 μm, smaller regular formations are visible. This is confirmed by a consideration of surface morphology at a scan area of 6 × 6 µm (Figure 1b). Mesoscale pores were identified, the structure of which is visualized in the 912 × 912 nm image (Figure 1c).

A section of the porous silicon structure 912 × 912 nm in size was scanned separately with a resolution of 512 × 512 pixels. The structures oriented at an angle to the axes of an imaginary horizontal plane excluding clearly defined boundaries are observed here. This indicates the presence of both amorphous and crystalline phases. The longitudinal dimensions of these structures are in the range of 80 to 115 nm, the transverse dimensions are 37–64 nm.

Figure 2 represents an image of the relief section of the porous silicon surface (por-Si) with an average roughness of 14.9 nm. It can be seen that the relief of the surface is characterized by the presence of single ridges and numerous hollows with a depth of 60 ± 10 nm and pores do not have a preferred orientation. The boundaries of the etching areas are varied. The total area occupied by hollow is 26.2% (Figure 2).

When depositing 20 layers of coating, optical microscopy depicts a decrease in the size of macropores. Regular formations are additionally visible in the 25 × 25 μm image. Zooming in to 1.5 by 1.5 µm showed that the formations basically have the same spatial orientation. An increase of 500 × 500 nm made it possible to identify ZnO nanoclusters (Figure 3).

The formation of mesoscale pores and clusters was identified on the sample surface after depositing 25 layers of coating, at a magnification of 500 × 500 nm (Figure 4). Their boundaries comprise a structure of gradual transition from formations of one type to another. A characteristic feature, in this case, is the presence of dangling bonds with an uncompensated charge.

Differences in the nature of porous silicon texture and the porosity presence of several levels affect the growth mechanism, morphology, phase formation, and other properties of the phase synthesized in pores.

Possible mechanisms of zinc oxide film growth are schematically shown in Figure 5.

In the process of film deposition on a heated substrate (400 °C), the air in the pores expands and ridges and hollows are formed on the surface of the sample (Figure 5b). This explains the formation of clusters when depositing 20 layers of coating. An increase in the number of layers to 25 leads to cluster enlargement and the hollows formation between them. The clusters are considered to be the points of coverage growth.

The average size of the crystallites for a sample with 25 ZnO layers was calculated from the half-width of the X-ray lines using the Debye–Scherrer expression [39].
D = 0.9 λ/B cosθ(1)
where λ is the X-ray wavelength, θ is the Bragg diffraction angle, and B is the full width at half maximum of a respective line. The calculation results are presented in Table 1.

Based on the fact that the coating was deposited in air atmosphere and on a hot substrate, the structure of the sample contains a thin layer (≈5 nm, according to ellipsometry) of silicon oxide.

Therefore, a composite system, which includes silicon, silicon oxide, and ZnO, was formed.

The surface morphology of a silicon sample is modified in various ways during the electrochemical etching process and after film deposition. Measurement of the specific surface area by the method of physical gas adsorption (BET) allows to determine these transformations.

### 3.2. Dependence of the Specific Surface Area of the Samples on the Number of Deposited Coating Layers

For the initial sample (por-Si), the specific surface area is about 0.87 m^2^/g, and for the sample with 20 coating layers it equals to 0.54 m^2^/g. This parameter is not significantly different for samples with 20 and 25 layers of ZnO coating.

As a result of the investigation, it was revealed that when depositing the film, the specific surface area decreases, which is due to the fact that the film covers the pores.

The higher the roughness of the sample surface, the more chemically active the particles are with dangling bonds—paramagnetic centers are on it. Their identification by the EPR method establishes a complete description of the characteristics.

### 3.3. EPR Studies

#### 3.3.1. General Investigations

EPR study of the surface-oriented structures has been carried out on the samples of initial porous silicon and por-Si with layers of ZnO.

EPR spectra were recorded using JES-FA spectrometer (Jeol) at 300 K temperature in the presence of standard Mn^2+^ in MgO, which causes EPR spectra with hyperfine structure (HFS) of 6 narrow lines due to influence of ^54^Mn nuclear spin I = 5/2 on unpaired electron.

The studies were based on the assumption of the electronic structure defects presence associated with layers formation and at the boundaries of pores and nanocrystals.

Registration conditions: Frequency 9.445 GHz, in the field Fc = 340 mT, sweep width 7.50 (mT), modulation frequency and amplitude Fr. = 100 (kHz), width = 0.6 (mT).

In the investigated structures, intense signals, which indicate the presence of high concentration of paramagnetic centers (PMCs), were recorded.

From the spectrum of pure porous silicon, in the interval between the 3rd and 4th lines of the accompanying standard, a signal with axial symmetry was registered, 1.6 mT wide and g-factor of 1.98 (Figure 6).

Its origin is associated with dangling bonds of the Si-O-O type. The concentration of PMCs of the order of 10^16^ is explained by disordered structures. 

For coated samples, the signal is similar to a singlet, and asymmetry of the signal is also observed (the area under the curve in the lower and upper parts is not the same) (Figure 6). The signal width is 1.37 G, the g- factor is ~ 1.987, which corresponds to the oxygen vacancy—Vo [40]. PMC localization occurs efficiently after coating deposition, and the signal enhances with an increase in the number of layers. This is presumably connected to the growth of ZnO clusters, i.e., the appearance of more ordered structures. 

The results of measuring the thickness of the ZnO coating by ellipsometry: 20 layers of ZnO—16 nm, 25 layers of ZnO—25 nm.

Indeed, it has been established that meso and macropores are formed on the silicon surface by electrochemical anode etching. Deposition of 20 and 25 layers of zinc oxide coating forms ZnO clusters as prototypes of crystalline inclusions.

The disorientation of crystal grains is associated with the formation of defects at their boundaries. This affects the properties of PMCs, similar to the formation of defects in SiO_2_ contained in natural solids, with an etched surface or under irradiation [41]. The identification of such structures is possible using the EPR signal saturation mode, which allows to determine even a weak anisotropy of the crystal lattice [41,42].

#### 3.3.2. Influence of EPR Signal Saturation on the Shape of Spectra

The change in signals with an increase in the power of microwave radiation in the range from 0.4 to 7.6 mW was investigated. A non-standard algorithm was opted for it. In several cycles, the power value was varying with an equal step of 0.8 mW in different intervals up to 7.6 mW.

The resulting EPR signals are in Figure 7, where the changes in the signal shape with increasing microwave power are illustrated (for a sample with 25 coating layers).

At low microwave power values, the signal shape with signs of anisotropy appears and persists with its growth up to 2.8 mW. This generally reflects a uniform distribution of identical PM centers in the sample. With a further increase in the power, the signs of signal asymmetry turn to be more pronounced.

This effect is due to the PMC anisotropy, i.e., the presence of several types of paramagnetic center as EPR signal source. It manifests itself in a more complex signal shape in the negative part of the spectrum with increasing power (Figure 7a–c).

The transformation of the PMC signal occurs according to the mechanism of homogeneous saturation. It can be concluded that the centers are localized uniformly in the volume of the sample. The structure of the centers is determined by axial asymmetry.

As the field power increases to 3.6 mW (Figure 7b), the component in the lower part of the spectrum continues to broaden (growth in the microwave power from 2.8 to 3.6 mW). Moreover, the lateral component in the complex signal turns to be more pronounced.

In Figure 7c, the lower and upper asymmetry parts of the spectrum are clearly visible (the difference in the derivative branches). In the lower part, with an increase in the power, a complex (anisotropic) signal associated with the presence of O_2_^−^ particles is clearly manifested.

Figure 7d shows a simulated spectrum containing signals from several paramagnetic centers. The lines, which do not reflect the useful signal, were averaged.

The saturation process in EPR spectroscopy is represented in terms of PMC signal/square root of microwave power, based on the dependence of power on current. The Figure 8 illustrates this dependence for samples with 20 and 25 coating layers.

In the range of low power values, the change in signal of the sample with 20 ZnO layers is close to linear. Nevertheless, as the power increases, the signal amplification decreases and an inflection is observed due to the saturation of spin transitions from the spin reservoir of the system to the upper level. However, with a further growth in the microwave power, a decrease in the signal is not observed and is replaced by an increase again, since then the reaction of another “spin-packet” is switched on, which requires the absorption of more energy to flip the spins. Thus, the presence of one more type of PMCs has been substantiated.

For sample with 25 ZnO layers, the signal intensity is higher, as the microwave power increases, the line shape remains stable whilst the amplitude increases. Consequently, there is a slower rise (above 4 mW) and a decay, thus the contribution of another type of centers is noticeable as the signal increases.

Thus, this analysis allowed to determine the anisotropy of the EPR signal and the presence of several types of paramagnetic centers in the investigated sample composition.

For the sample with 25 layers, the PMCs concentration is higher, the saturation peak is shifted, and the inflection is observed at a higher power value, therefore, PMC localization is more ordered. Based on this, PMCs exchange energy more in the form of thermal collisions, losing energy. The relaxation time of this type of PMC is shorter for a sample with 25 layers, since it quickly releases energy in the form of oscillations and passes from the excited to the ground state. These statements are a sign of the formation of the crystalline structure. This is also due to the mode of deposition films to the substrate at 350–400 °C, at which a polycrystalline surface structure is formed.

The EPR spectra are observed in the form of the first derivative. The inverse operation–integration, allows to estimate the area under the curve of the spectrum, its shape, in addition to decompose the spectrum into components.

With an increase in the applied power, the signal intensity increases while the concentration of the active paramagnetic centers increases (Figure 9a).

It can be seen in Figure 9b that the spectrum is decomposed into two Gaussians. This confirms the assumption of two types of paramagnetic particle existence, which are responsible for the EPR signal in this sample. Signal approximation parameters are denoted in Table 2.

Based on the fact that the fundamental difference between PMCs is the variety of relaxation times, further studies of EPR signal saturation in a wider than 0.8–7.6 mW range of microwave power using spectrum integration and decomposition into components is a promising research area.

Identified defects can be of different nature. When being excited, some of them can recombine emitting quants of light.

### 3.4. Photoluminescence Studies

Due to the presence of a large number of luminescence models for porous silicon and zinc oxide, research in this direction is very relevant [43,44,45]. 

In the process of transforming the morphology of the samples surface, the sizes of structures and their orientation in crystal lattice vary. The purpose of this section is to determine the properties of light-emitting particles with these changes.

Figure 10 represents the spectra of the samples before and after the film deposition.

PL spectra of the samples were measured at room temperature with excitation at a wavelength of 320 nm. The spectrum of porous silicon disappears after the film deposition, which is associated with the passivation of the surface with a ZnO coating layer. For a sample with a ZnO film, two PL bands are present: a high-intensity green emission band centered at 507 nm (2.43 eV) and a weak-intensity UV emission peak with a center at 385 nm (3.22 eV).

It is possible to attribute the visual emissions detected to the intrinsic defects and oxygen vacancies [46,47,48], although the exact mechanism responsible for these emissions is still in debate.

It is noticeable that with an augmentation in the number of deposited layers, the signal intensity at 380 nm rises, which is associated with an increase in the film thickness.

#### 3.4.1. Photoluminescence of Porous Silicon

As a rule, the PL spectrum of porous silicon is a broad (∆λ ~ 100–200 nm) structureless band, the maximum position of which can shift from 750 nm (red light) to 450 nm (blue light) with a change in the electrochemical treatment modes. The radiation is formed in nanostructures (fibers and clusters) of silicon, the electronic spectrum of which is modified due to the quantum confinement effect [49,50].

At a low defect concentration in a bulk porous silicon crystal, two emission mechanisms are possible: direct radiative recombination of a free electron and hole or exciton annihilation. The binding energy of carriers in an exciton at room temperature is very low due to efficient thermal dissociation. This, along with the indirect structure of silicon energy bands, explains the low value of the PL quantum yield.

Thus, already at room temperature, photoexcited charge carriers in porous silicon form two dynamically coupled subsystems: free electrons, holes, and excitons.

#### 3.4.2. Photoluminescence of Zinc Oxide

All characterized ZnO thin films comprise two bands of PL. The first is centered at 380 nm (ultraviolet) photoluminescence near the intrinsic absorption edge (edge luminescence), and the second is about 520 nm wide long-wavelength band, its maximum falls in the green part of the spectrum, which occurs during radiative transitions through deep levels. The spectral position of the band at 380 nm (3.27 eV) is attributed to exciton recombination at the near-band boundary in ZnO films, while the blue-green emission band centered at 520 nm (2.38 eV) is attributed to the non-stoichiometric composition of ZnO (defects, mainly oxygen vacancies). A single ionized oxygen vacancy is the most frequently cited hypothesis for the origin of green emission in ZnO.

There are three charge states for oxygen vacancies in ZnO: 0, +1, and +2, which correspond to $V_{o}^{0}$, $V_{o}^{+}$, and $V_{o}^{2+}$ respectively. $V_{o}^{0}$ and $V_{o}^{2+}$ centers are thermally stable, whereas $V_{o}^{+}$ centers are unstable and transform into V_O_ centers, capturing electrons from the conduction band.

Small (~100 nm) granules in the film cause the luminescence band to be recorded at 2.2 eV (~555 nm), and the luminescence centers are oxygen vacancies $V_{o}^{2+}$, which emit when electrons are captured from the conduction band (Figure 11a).

Vacancies in the form of $V_{o}^{2+}$ prevail at the grain boundaries, while vacancies in the form of $V_{o}^{0}$ predominate in the bulk. The level of the $V_{o}^{0}$ vacancy is located 0.86 eV below the bottom of the conduction band, and the level of the $V_{o}^{2+}$ vacancy is 1.16 eV above the top of the valence band. Similar processes are typical for nanocrystals: two main luminescence bands were recorded with maxima at 506 nm, which is responsible for hole recombination at $V_{o}^{+}$ centers in the bulk of the sample, and at 564 nm, associated with electronic transitions at $V_{o}^{2+}$ centers in the depleted region of the sample.

Under the action of UV radiation, a neutral oxygen vacancy transforms into an excited singlet state (S = 0), then it relaxes into an excited triplet (S = 1) state, which results in the radiation of the center (Figure 11c).

In nanoparticles, neutral oxygen vacancies are formed near the surface. Therefore, they are characterized by surface green luminescence. In samples with an excess of zinc-oxygen vacancies serve as luminescence centers, and with an excess of oxygen, zinc vacancies. Naturally, Oi interstitial oxygen centers are also formed in O-excess samples, while interstitial zinc centers are formed in Zn-excess samples. These centers create shallow donor levels in band gap. In the general case, a wide ZnO emission band in the visible region of the spectrum can consist of two or more bands associated with different luminescence centers.

## 4. Conclusions

Thus, it has been established that macropores, including mesopores, are formed on the silicon surface by electrochemical anode etching. Depositing of 20 layers of zinc oxide coating forms ZnO nanocrystals. With an increase in the number of layers to 25, the nanocrystals get enlarged and hollows are formed between them. The distribution of these formations over the surface of the sample and their size are identical. The mechanism of structure formation is determined, where an essential role is identified by the fact that the films are deposited on a hot substrate. In the EPR spectrum of porous silicon (without coating), a signal was recorded with signs of axial symmetry, the origin of which is associated with dangling bonds of the Si–O–O type. The PMCs concentration is approximately 10^16^, which is explained by the disordered structure of the initial sample. With an increase in the number of deposited layers, the EPR signal enhances. Several reasons for this have been identified; however, the main contribution is made by oxygen vacancies. The investigation of the EPR saturation signal demonstrated that the sample with 25 layers contains PMCs of the same characteristics, uniformly distributed in the volume of the sample. The localization of PMCs turns to be more ordered with an augmentation in the number of layers. The relaxation time of centers decreases with an increase in the number of layers, which is associated with the formation of energetically stable particles in the volume of ZnO nanoclusters. In the photoluminescence investigation of the initial sample, a typical band of porous silicon with low-intensity was registered. With an increase in the number of deposited ZnO layers, the PL rises. The exciton glow emission is maximum when 25 ZnO layers are formed, which is associated with an increase in the film thickness. The greatest intensity of PL is associated with the presence of oxygen vacancies in the structure of samples.

## Figures and Tables

**Figure 1 materials-16-00838-f001:**
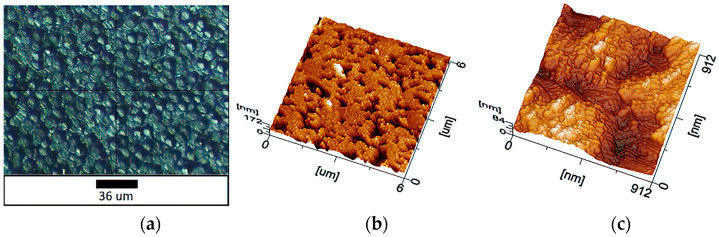
AFM images of the sample without coating: (**a**) optical microscopy; (**b**) 6 × 6 μm (topography mode); (**c**) 912 × 912 nm (topography mode).

**Figure 2 materials-16-00838-f002:**
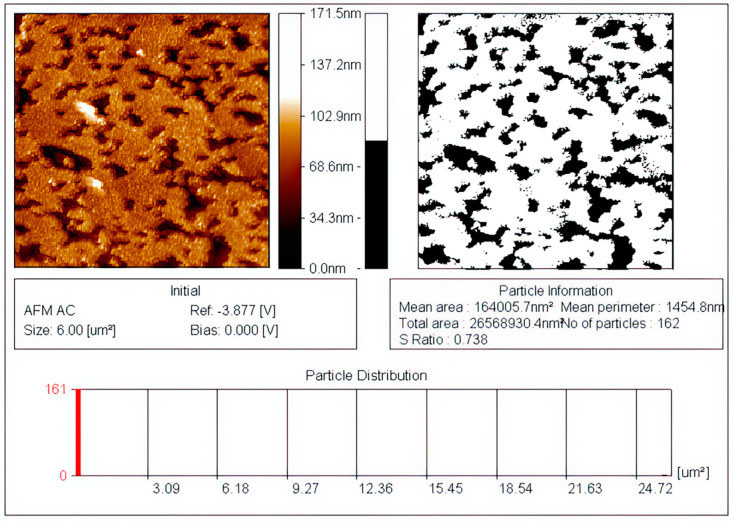
Detailed information of AFM images of an uncoated sample: the ratio of the area occupied by pores to the total surface area of the sample.

**Figure 3 materials-16-00838-f003:**
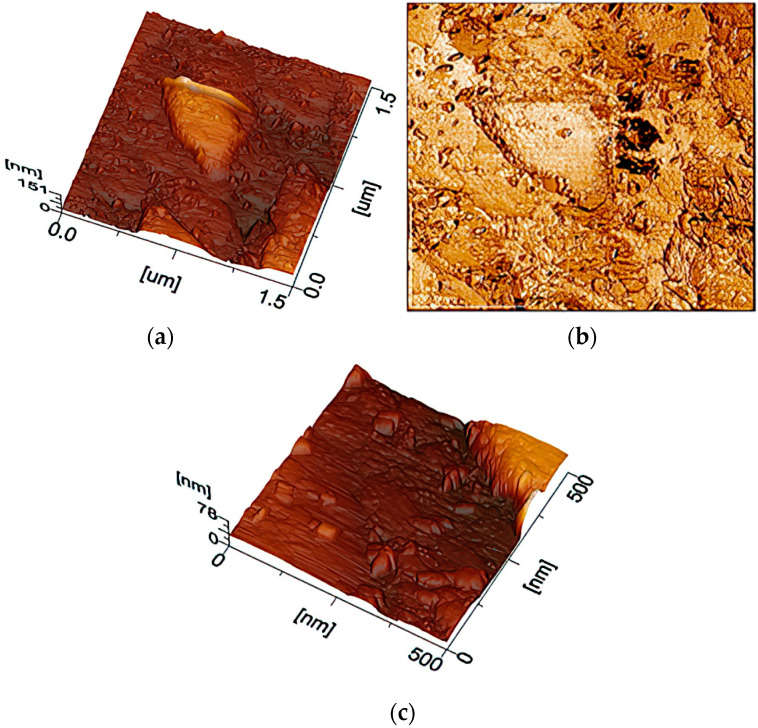
AFM images of a sample with 20 coating layers, scan scale—1.5 × 1.5 µm: (**a**) topography mode; (**b**) phase mode; (**c**) topography mode.

**Figure 4 materials-16-00838-f004:**
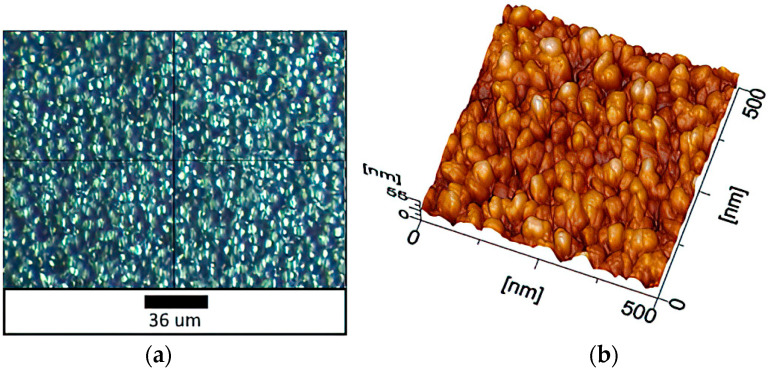
AFM images of a sample with 25 coating layers: (**a**) optical microscopy; (**b**) 500 × 500 nm (topography mode).

**Figure 5 materials-16-00838-f005:**
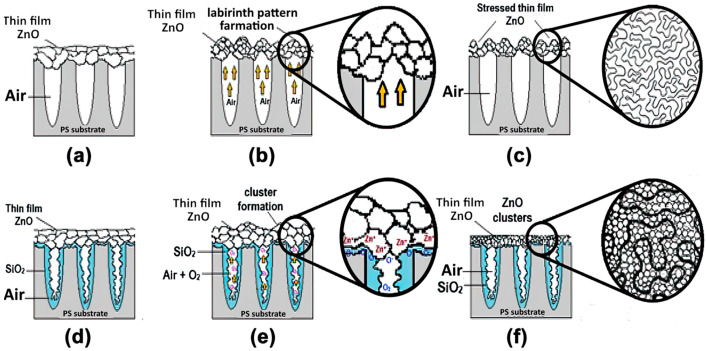
Mechanisms of zinc oxide film growth on the surface of porous silicon [38]: (**a**–**c**) Stages of the formation of granular labyrinth pattern on SiO_2_/Si substrate; (**d**–**f**) stages of the deposition of ZnO on SiO_2_/Si substrate with small pores.

**Figure 6 materials-16-00838-f006:**
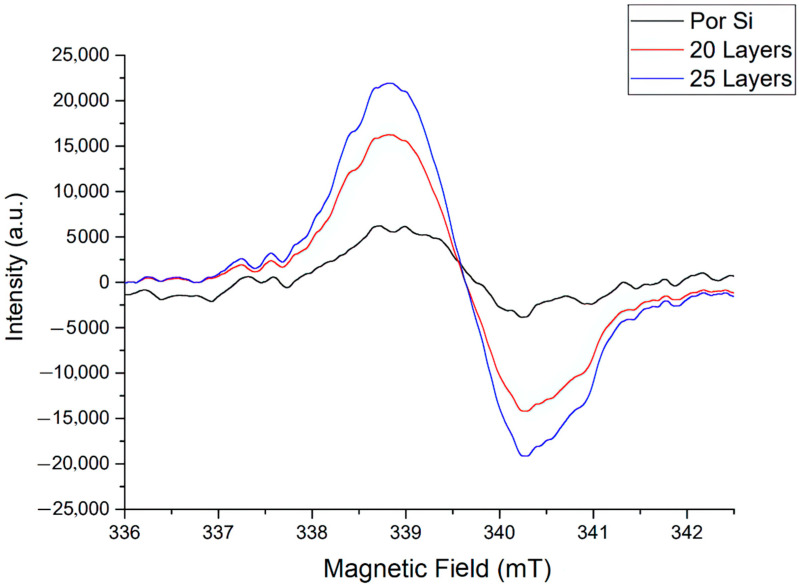
EPR spectra.

**Figure 7 materials-16-00838-f007:**
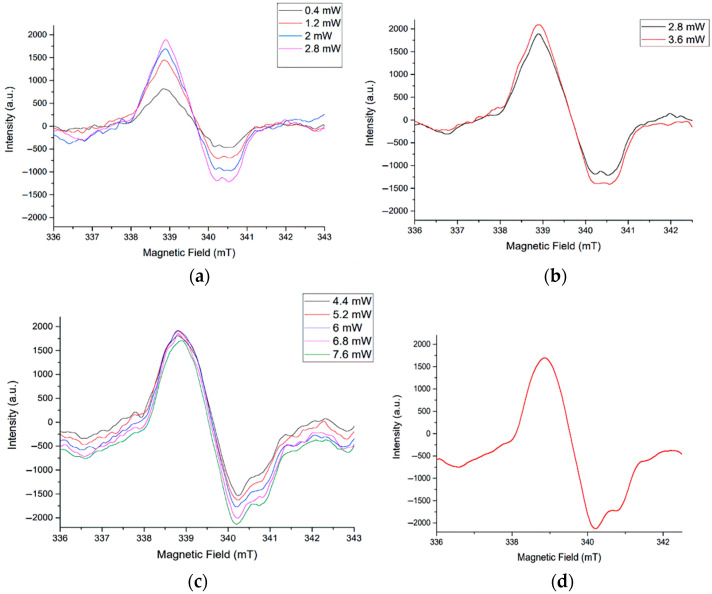
EPR signal saturation spectra for a sample with 25 layers of ZnO coating: (**a**) Changing of the microwave power from 0.4 to 2.8 mW; (**b**) changing of the microwave power from 2.8 to 3.6 mW; (**c**) changing of the microwave power from 4.4 to 7.6 mW; (**d**) averaged EPR spectrum at signal saturation.

**Figure 8 materials-16-00838-f008:**
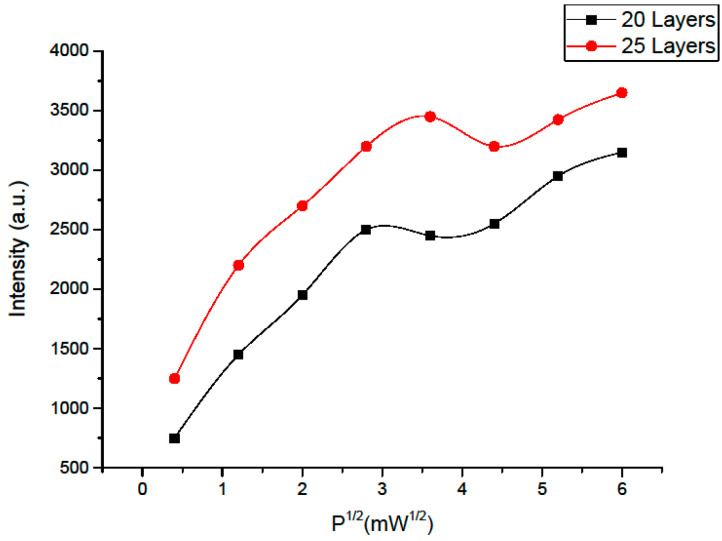
Dependence of the EPR signal intensity on P^1/2^.

**Figure 9 materials-16-00838-f009:**
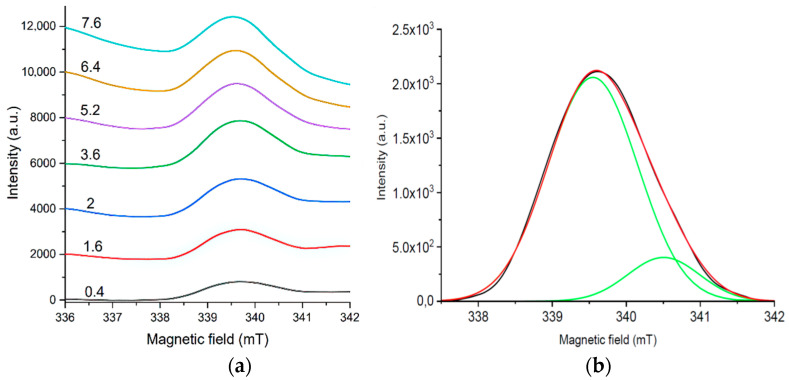
Integrated spectra of the EPR signal saturation for a sample with 25 layers of ZnO coating: (**a**) The power of magnetic field varies from 4.4 to 7.6 mW; (**b**) decomposition into Gaussians of the spectrum at a magnetic field power of 7.6 mW.

**Figure 10 materials-16-00838-f010:**
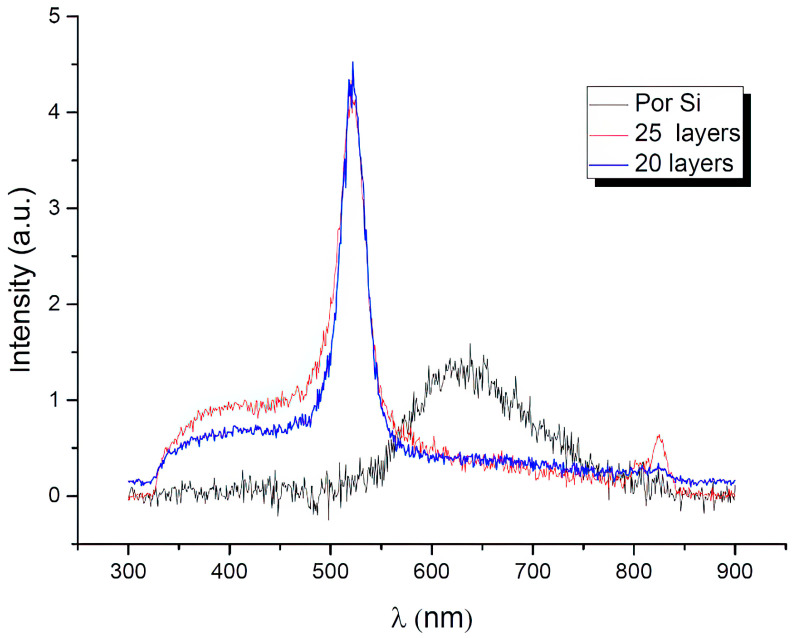
PL spectra of the initial sample and with ZnO coating.

**Figure 11 materials-16-00838-f011:**
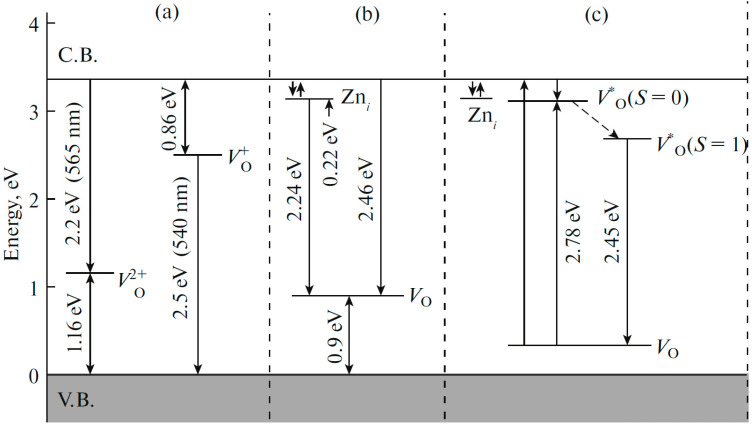
Energy level diagram of ZnO and possible transitions responsible for green luminescence (GL) [51]: (**a**) Luminescence model for films consisting of large or small granules; (**b**) luminescence model for nanoneedles and polycrystals; (**c**) a model of triplet-singlet transitions of $V_{o}^{0}$ centers as explanation of the GL [52,53,54,55,56,57,58,59].

**Table 1 materials-16-00838-t001:** Average size of ZnO crystallites in films.

Average Size of Crystallites in the Plane, nm		
ZnO (100)	ZnO (002)	ZnO (101)
15.4	14.6	13.1

**Table 2 materials-16-00838-t002:** Approximation parameters of the EPR signal for a sample with 25 coating layers, when the spectrum is decomposed into components.

Peak Index	Peak Type	Peak Area by Integrating Data	FWHM	Max Height	Peak Gravity Center (mT)	Peak Area by Integrating Data (%)
1	Gauss	3411.39144	1.50414	2132.40004	339.6	94.10617
2	Gauss	213.6539	0.88195	228.48705	340.62942	5.89383

## Data Availability

Not applicable.

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
