# Peer review of "Self-Organization Effects of Thin ZnO Layers on the Surface of Porous Silicon by Formation of Energetically Stable Nanostructures"

_materials, 2023, doi:10.3390/ma16020838_

Round 1

Reviewer 1 Report

The authors present properties of ZnO prepared on porous silicon. The overall results is presented clearly however some improvements are required to improve the manuscript:

1. I don't see the problem statement that the authors would like to tackle. It is not clear in the introduction section. There should be a comparison to other works related to the solution and how the authors provide some form of novelty to this. 

2. Authors need to define what EPR and PMC is before using the short form.

3. The authors claim the novelty is the formation of paramagnetic centres through their method of sample preparation. The methods mentioned are not new. There are many articles using the same method to form ZnO on porous silicon. This statement must be revised. I believe the novelty is in identifying the paramagnetic centres of this material deposited. 

4. It is not clear that the material on the surface is ZnO. The authors should identify what this material is perhaps EDX measurement to identify the % of elements at the surface? Maybe XRD measurement to show that ZnO formation is detected? There must be some means of identifying what exactly this material is. 

5. The PL spectra should show narrow peak related to ZnO at 380 nm. However the authors show a very broad peak at 380 nm and a very narrow peak at 520 nm. This is not conventionally observed. It is usually the other way round. Is the source different for generating the 380 nm peak and the 520 nm peak? The authors must clarify this observation. The authors must also link this observation with PMC. 

6. Overall, a comparison with other literature must be provided on the structure formed (how different is it with other reported work?), the PMC identified (how do they vary with other works?) etc. 

7. English editing is required through out the manuscript. 

Author Response

  1. I don't see the problem statement that the authors would like to tackle. It is not clear in the introduction section. There should be a comparison to other works related to the solution and how the authors provide some form of novelty to this. 

Thank you for pointing this out, we have corrected the introduction taking into account the problem statement:

The process of forming materials with high luminescent and sensory properties is of great interest. The nucleation of structures occurs at the level of interaction of individual particles, with an uncompensated charge. However, the isolation of the photoluminescense and EPR spectroscopic signal of such particles is an extremely difficult task.

Nanocrystalline structures have high luminescent characteristics and sensitivity when reacting with gases. Nanocrystal–amorphous substance transitions have the structure of a gradual transition from formations of one type to another and therefore contain particles with dangling bonds that differ in structure. The key to understanding the mechanism of transition is related to the nature of the paramagnetic centers (PMCs) interaction.

During the formation of PMCs their structure and energy properties change. The transformation of the PMC is limited in time, that is, one of the main characteristics is the relaxation time of the particles. Therefore, the study of the EPR signal saturation, based on the dependence of the properties of the PMC on the relaxation time, will allow us to identify the process of nucleation of nanostructures and the mechanism of their formation.

  1. Authors need to define what EPR and PMC is before using the short form.

Thanks to the reviewer for this comment. We defined the abbreviations before using the short form.

  1. The authors claim the novelty is the formation of paramagnetic centres through their method of sample preparation. The methods mentioned are not new. There are many articles using the same method to form ZnO on porous silicon. This statement must be revised. I believe the novelty is in identifying the paramagnetic centres of this material deposited.

Thank you. We have adjusted the novelty of the research taking into account the remark:

The novelty of this study is to determine the process of nanoclusters in por-Si/ZnO structures formation by the EPR signal saturation method, based on the dependence on the relaxation time of the paramagnetic center.

  1. It is not clear that the material on the surface is ZnO. The authors should identify what this material is perhaps EDX measurement to identify the % of elements at the surface? Maybe XRD measurement to show that ZnO formation is detected? There must be some means of identifying what exactly this material is.

Thank you for this excellent observation. We have inserted the XRD section into the article:

Investigation of the structure of ZnO thin films

The structure of the deposited films was studied by X-ray diffraction (XRD) with the DRON-6 (LOMO, Russia) setup. A narrow-collimated (0.05x1.5) mm2 of a monochromatic (CuKa) X-ray beam directed at an angle of 5° to the surface of the sample was used. The intensity of X-ray reflections along the debagram was measured every 2θ=0.05 on the MD-100 microdensitometer.

The average size of the crystallites for a sample with 25 ZnO layers was calculated from the half-width of the X-ray lines using the Debye-Scherrer expression [39].

D=0.9 λ/B cosθ,

where λ is the X-ray wavelength, θ is the Bragg diffraction angle and B is the full width at half maximum of a respective line.

Table 1. Average size of ZnO crystallites in films

Average size of crystallites in the plane, nm

ZnO (100)

ZnO (002)

ZnO (101)

15,4

14,6

13,1

  1. The PL spectra should show narrow peak related to ZnO at 380 nm. However the authors show a very broad peak at 380 nm and a very narrow peak at 520 nm. This is not conventionally observed. It is usually the other way round. Is the source different for generating the 380 nm peak and the 520 nm peak? The authors must clarify this observation. The authors must also link this observation with PMC. 

Thank you for pointing this out. We have found out the nature of the PL peaks, but there are still points for discussion.

For a sample with a ZnO film two PL bands are present: a high-intensity green emission band centered at 507 nm (2.43 eV) and a weak-intensity UV emission peak with a center at 385 nm (3.22 eV).

It is possible to attribute the visual emissions detected to the intrinsic defects and oxygen vacancies, although the exact mechanism responsible for these emissions is still in debate.

  1. Overall, a comparison with other literature must be provided on the structure formed (how different is it with other reported work?), the PMC identified (how do they vary with other works?) etc.

Thanks to the reviewer for this comment. We have carefully studied other works in this area of research and have come to the following conclusion:

In [15], one of the frequently observed EPR signals in ZnO by an isotropic g-factor of ~1,955–1,964 was identified. This PMC signal, according to the opinion established in the literature, is attributed to a single-charge oxygen vacancy, as analogous to F-centers.

Earlier, when studying ZnO in a high field, an EPR signal with axial anisotropy of the g-factor was identified, which, according to the authors [5], indicates the presence of several types of particles with uncompensated charge in the structure.

In our work, signals with signs of axial asymmetry are recorded in the original pure porous silicon subjected to heat treatment in the presence of residual oxygen. We believe that, apparently, this is a signal of surface states of the type of dangling hybrid s - p orbital that participate in interactions with the medium [6,7].

This makes it possible for us to identify the observed signals in porous silicon coated with ZnO, the shape of which suggests a superposition of two signals broadened due to anisotropic interactions in disordered fragments of the coating and dipolar spreading. The type of saturation curve confirms this conclusion even more clearly, the results are consistent with the conclusions of the study of luminescence.

Presumably, the PMC signals are associated with oxygen vacancies in ZnO with an admixture of a residual weak signal of surface bonds with dangling bonds of the Si – O – O type.

With an increase in the number of layers, the signal not only grows, but also noticeably narrows, with a shift in the region of a strong field, which confirms our interpretation and allows us to associate it presumably with the growth of clusters, the appearance of more ordered structures.

We should add that we have not specifically studied the anisotropy of the g–tensor, this can be done in future studies, as well as a more detailed study of the dependence of signal parameters on the number of surface ZnO layers.

In [8], magnetic anisotropy in monocrystalline ZnO weakly doped with cobalt was investigated, especially due to their inverse and forward piezoelectric properties (PZ) in combination with their n-type semiconductor character.

The authors report that Saturation is characterized by an intensity proportional to √ P occurred at a very low power level, the unsaturating case is very low (0.5 MW). When studying microcrystals at a low temperature (about 5 K), it is not possible to obtain a detectable signal leading to a saturated mode due to the number of rotations, it is weak, so high incident microwave power is often required to obtain a result. Therefore, the study of saturation was supported by the study of the temperature dependence of the signal.

In our case, we are dealing with polycrystalline samples, however, anisotropy effects may be present due to the orientation of the PMC in local fragments of growing and forming surface layers.

The use of the magnetic anisotropy of a coating with a dopant can be useful for our further studies of the kinetics of layer growth. As well as the study of the relationship of the parallel and perpendicular components of an anisotropic signal. Of these magnetic anisotropies to characterize the crystal quality of microcrystals with a low content of doped ZnO: Co.

  1. Savchenko, D.; Vasin, A.; Kuz, O.; Verovsky, I.; Prokhorov, A.; Nazarov, A.; Lančok, J.; Kalabukhova, E. Role of the paramagnetic donor‑like defects in the high n‑type conductivity of the hydrogenated ZnO microparticles. Nature 2020, 10, 1-9. https://doi.org/10.1038/s41598-020-74449-3 (in Paper reference [19])
  2. Mochalov, L.; Logunov, A.; Prokhorov, I. et al. Variety of ZnO nanostructured materials prepared by PECVD Opt Quant Electron. 2022, 54, 1- 4. https://doi.org/ 10.1007/s11082-022-03979-z (in Paper reference [5])
  3. Warren, W.L.; Lenahan, P.M.; Curry, S.E. First observation of paramagnetic nitrogen dangling-bond centers in silicon nitride. Physical review letters 1990, 65, 207-210.
  4. Morrison, S.R. The Chemical Physics of Surfaces. Stanford Research Institute. Plenum Press – NY, London. 1977 (in Russian). Mir. Moskva. 1980. 488P.
  5. Savoyant, A.; Pilone, O.; Bertaina, S.; Delorme, F.; Giovannelli, F. Electronic and nuclear magnetic anisotropy of cobalt-doped ZnO single-crystalline microwires. Superlattices and Microstructures 2019, 125, 113-119. https://doi.org/ 10.1016/j.spmi.2018.10.024 (in Paper reference [42])

  1. English editing is required through out the manuscript. 

Thank you for pointing out this important issue. We have edited the manuscript.

Reviewer 2 Report

Manuscript ID: catalysts-1976912

I have read the manuscript entitled “ZnO thin layers self-organization effects on the porous silicon surface by formation energetically stable nanostructures”. In this article the authors have described the film deposition by spray pyrolysis. They have claimed that the ZnO nanoclusters regularly distributed over the sample surface. However, the authors failed to explain and draw out the novelty of the work, this aspect needs to be improved. This work is worthwhile to be publish in this journal after minor revisions. The following issues should be addressed:

Please rewrite the abstract with including the statement of problem.

1.      Introduction part is very short and can be improved by citing more recent work in the field.

2.      Add the complete details of the chemicals and reagents with their source separately.

3.      Write complete information of the instruments used (model, country etc).

4.      The conditions used during EPR research are self-selected or just repeating of the previous work.

5.      How the current research is better than previous reported works.

6.      There is no identification of the synthesized materials via important techniques like EDX and FTIR. Must be included.

7.      Why materials other than ZnO was not used for deposition on silicon (like TiO2).

8.      Kindly add a table for comparing the current study with previous research.

9.      “With an increase in the number of deposited ZnO layers the PL augments. The exciton glow emission is maximum when 25 ZnO layers are formed, which is associated with an increase in the film thickness”. What about for the layers more than 25. Why the authors did not study the optimum value of ZnO layers.

10.  Write future prospects of the current study in 2-3 lines.

11.  The importance and novelty of the research should be highlighted and more clearly stated.

12.  English of the manuscript needs substantial improvement.

Author Response

2 reviewer

Please rewrite the abstract with including the statement of problem.

Thanks to the reviewer for this comment. We have edited the abstract taking into account the problem statement:

The formation of complex surface morphology of a multilayer structure, the processes of which are based on quantum phenomena, is a promising domain of ​​the research. A hierarchy of pore of various sizes was determined in the initial sample of porous silicon by the atomic force microscopy. After film deposition by spray pyrolysis, ZnO nanoclusters regularly distributed over the sample surface were formed. Using the electron paramagnetic resonance (EPR) method it was determined that the localization of paramagnetic centers occurs more efficiently as a result the ZnO deposition. An increase in the number of deposited layers, leads to a decrease in the paramagnetic center relaxation time, which is probably connected with the formation of ZnO nanocrystals with energetically stable properties. The nucleation and formation of nanocrystals is associated with the interaction of particles with an uncompensated charge. There is no single approach to determine the mechanism of this process. By the EPR method supplemented with the signal cyclic saturation, spectral manifestations from individual centers were effectively separated. Based on electron paramagnetic resonance and photoluminescense studies it was revealed that, the main transitions between energy levels are due to oxygen vacancies and excitons.

  1. Introduction part is very short and can be improved by citing more recent work in the field.

Thank you for pointing this out, we have supplemented the introduction with data from recent articles on this scientific direction.

  1. Add the complete details of the chemicals and reagents with their source separately.

Thank you for pointing out this important issue. We have updated the information about chemicals and reagents:

The sol-gel method was applied to prepare the film-forming solution. The sol solution was obtained from zinc acetate dihydrate (Zn(CH3COO)2·2H2O) with a concentration of 0.1 M, mixed with 9 ml isopropanol (C3H7OH) as a solvent with the addition of 1 ml monoethanolamine (C2H7NO) as a stabilizing agent. The solutions used were prepared on the LAB-PU-01 orbital rotation shaker (rotation speed 150 rpm). The dissolution was carried out both at room temperature and when heated to 50°C, which was a significant condition for the homogenization of the resulting solutions. The time of complete preparation of the solution was 1 hour. Deposition of ZnO films on the substrate began no later than 1 hour after the preparation of solutions.

  1. Write complete information of the instruments used (model, country etc).

Thank you so much for catching these errors. We have updated the information about the models of devices and the countries of the manufacturer.

  1. The conditions used during EPR research are self-selected or just repeating of the previous work.

Thank you for pointing out this important issue.

In our work, we selected a scan center in the g-factor region corresponding to electronic centers, the scan width allowed us to navigate by 3-4 components of the Mn (+2) /MgO ultrathin structure. The same approach was applied to adjust the frequency and amplitude of the modulation. Since the signal width at registration at 293K is large enough, the average modulation amplitude is selected accordingly to avoid additional broadening. We didn't need to change the modulation frequency, as we were satisfied with the ratio of selectivity and sensitivity. The time parameters also satisfy the task.

Under the selected conditions, the effect of changes in microwave power is clearly shown.

To separate and identify EPR signals during their saturation, a new technique based on a cyclic research process was applied. The change in signals with an increase in the power of microwave radiation in the range from 0.4 to 7.6 mW was investigated. A non-standard algorithm was opted for it. In several cycles the power value was varying with an equal step of 0.8 mW in different intervals up to 7.6 mW.

  1. How the current research is better than previous reported works.

Thanks to the reviewer for this comment.

In previous works, EPR spectroscopy of por-Si/ZnO structures was also studied. The advantage of our work is a cyclic consideration of the saturation of the EPR signal. This new technique with integrating the spectrum and decomposing it into components allowed us to isolate the signal from individual paramagnetic centers.

The novelty of this study lies in the determination of the process of nanoclusters formation in por-Si/ZnO structures by the EPR signal saturation method, based on the dependence on the relaxation time of the paramagnetic center.

  1. There is no identification of the synthesized materials via important techniques like EDX and FTIR. Must be included.

Thanks to the reviewer for this comment. We have inserted the XRD section into the article.

Investigation of the structure of ZnO thin films

The structure of the deposited films was studied by X-ray diffraction (XRD) with the DRON-6 (LOMO, Russia) setup. A narrow-collimated (0.05x1.5) mm2 of a monochromatic (CuKa) X-ray beam directed at an angle of 5° to the surface of the sample was used. The intensity of X-ray reflections along the debagram was measured every 2θ=0.05 on the MD-100 microdensitometer.

The average size of the crystallites for a sample with 25 ZnO layers was calculated from the half-width of the X-ray lines using the Debye-Scherrer expression [39].

D=0.9 λ/B cosθ,

where λ is the X-ray wavelength, θ is the Bragg diffraction angle and B is the full width at half maximum of a respective line.

Table 1. Average size of ZnO crystallites in films

Average size of crystallites in the plane, nm

ZnO (100)

ZnO (002)

ZnO (101)

15,4

14,6

13,1

  1. Why materials other than ZnO was not used for deposition on silicon (like TiO2).

Thank you for pointing out this important issue. We have made a comparison with different materials based on other works:

Due to their unique properties, II-VI group semiconductors are widely used in solar cells, field effect transistors, optoelectronic devices, diluted magnetic semiconductors (DMS), photoluminescence appliances [14–16]. Owing to a high exciton binding energy (60 meV) and a wide band gap (3.4 eV), ZnO is one of the most useful among them. Therefore, zinc oxide is one of the blue-ultraviolet (UV) region's most promising photonic materials.

  1. Kindly add a table for comparing the current study with previous research.

Thanks to the reviewer for this comment. We conducted a comparative analysis of the results of our work with previous works.

  1. “With an increase in the number of deposited ZnO layers the PL augments. The exciton glow emission is maximum when 25 ZnO layers are formed, which is associated with an increase in the film thickness”. What about for the layers more than 25. Why the authors did not study the optimum value of ZnO layers.

Thank you for pointing this out.

When selecting the number of coating layers, the goal was to obtain uniformly distributed nanocrystals on the sample surface. Microscopy results showed the presence of nanoclusters with a uniform distribution over the surface for the sample with 25 layers of ZnO. The EPR method proved that when depositing 20 and 25 layers, the same paramagnetic uniformly distributed in the sample volume are formed. This is possible only when identical structures are formed during the synthesis of the sample.

  1. Write future prospects of the current study in 2-3 lines.

Thanks to the reviewer for this comment. We have added information about the future prospects of the current study:

Based on the fact that the fundamental difference between PMCs is the variaty in relaxation times, further studies of EPR signal saturation in a wider than 0.8- 7.6 mW range of microwave power using spectrum integration and decomposition into components is a promising research direction.

The approaches developed in the work and the results obtained indicate the prospects for further research in this direction, which will allow us to understand the mechanisms and conditions for the formation of active layered structures on the surface of porous silicon.

  1. The importance and novelty of the research should be highlighted and more clearly stated.

Thank you for this excellent observation. We corrected the novelty of the research and described its importance.

The novelty of this study lies in the determination of the process of nanoclusters formation in por-Si/ZnO structures by the EPR signal saturation method, based on the dependence on the relaxation time of the paramagnetic center.

The importance of these studies lies in a new approach to studying the EPR signal saturation process by stepwise changing the microwave power values at certain variable intervals. This made it possible to more clearly represent the change in the shape and component of the signal. The representation of signals in integral form with decomposition into components is also used for the purpose of identifying signals of different PMCs.

  1. English of the manuscript needs substantial improvement.

Thanks to the reviewer for this comment. We have corrected the text of the article
